# K_V_ Channel-Interacting Proteins in the Neurological and Cardiovascular Systems: An Updated Review

**DOI:** 10.3390/cells12141894

**Published:** 2023-07-20

**Authors:** Le-Yi Wu, Yu-Juan Song, Cheng-Lin Zhang, Jie Liu

**Affiliations:** Department of Pathophysiology, Shenzhen University Medical School, Shenzhen 518060, China; 2200243053@email.szu.edu.cn (L.-Y.W.); 2060243049@email.szu.edu.cn (Y.-J.S.); liuj@szu.edu.cn (J.L.)

**Keywords:** K_V_ channel, K_V_ channel-interacting proteins, neurodegenerative disorders, cardiovascular diseases

## Abstract

K_V_ channel-interacting proteins (KChIP1-4) belong to a family of Ca^2+^-binding EF-hand proteins that are able to bind to the N-terminus of the K_V_4 channel α-subunits. KChIPs are predominantly expressed in the brain and heart, where they contribute to the maintenance of the excitability of neurons and cardiomyocytes by modulating the fast inactivating-K_V_4 currents. As the auxiliary subunit, KChIPs are critically involved in regulating the surface protein expression and gating properties of K_V_4 channels. Mechanistically, KChIP1, KChIP2, and KChIP3 promote the translocation of K_V_4 channels to the cell membrane, accelerate voltage-dependent activation, and slow the recovery rate of inactivation, which increases K_V_4 currents. By contrast, KChIP4 suppresses K_V_4 trafficking and eliminates the fast inactivation of K_V_4 currents. In the heart, *I*_Ks_, *I*_Ca,L_, and *I*_Na_ can also be regulated by KChIPs. *I*_Ca,L_ and *I*_Na_ are positively regulated by KChIP2, whereas *I*_Ks_ is negatively regulated by KChIP2. Interestingly, KChIP3 is also known as downstream regulatory element antagonist modulator (DREAM) because it can bind directly to the downstream regulatory element (DRE) on the promoters of target genes that are implicated in the regulation of pain, memory, endocrine, immune, and inflammatory reactions. In addition, all the KChIPs can act as transcription factors to repress the expression of genes involved in circadian regulation. Altered expression of KChIPs has been implicated in the pathogenesis of several neurological and cardiovascular diseases. For example, KChIP2 is decreased in failing hearts, while loss of KChIP2 leads to increased susceptibility to arrhythmias. KChIP3 is increased in Alzheimer’s disease and amyotrophic lateral sclerosis, but decreased in epilepsy and Huntington’s disease. In the present review, we summarize the progress of recent studies regarding the structural properties, physiological functions, and pathological roles of KChIPs in both health and disease. We also summarize the small-molecule compounds that regulate the function of KChIPs. This review will provide an overview and update of the regulatory mechanism of the KChIP family and the progress of targeted drug research as a reference for researchers in related fields.

## 1. Introduction

K_V_ channel-interacting proteins (KChIPs) are a family of Ca^2+^-binding EF-hand proteins consisting of four members: KChIP1, KChIP2, KChIP3 (DREAM/calsenilin), and KChIP4 (CALP), encoded by *KCNIP1-4*, respectively. KChIPs have high sequence similarity and share the same conserved C-terminal core domains. However, the N-termini of KChIPs are variable in sequence and length, which defines the distinctions among them. KChIP1, KChIP3, and KChIP4 are highly expressed in the brain, whereas KChIP2 is predominantly expressed in the heart [1]. Other tissues, such as lung [2], kidney [3], and gastrointestinal tract [4,5,6], also express low levels of KChIPs (Table 1).

The biological functions of KChIPs are diverse. In 1998, KChIP3, the first reported member of the KChIP family, was identified as a presenilin (PS)-binding protein and was initially named “calsenilin” [7]. A year later, the transcriptional regulatory activity of KChIP3 was reported by Carrion et al., as it can bind to the downstream regulatory element (DRE) on the promoter of the human prodynorphin (*PDYN)* gene, repressing *PDYN* transcription. KChIP3 is therefore coined the acronym DREAM (downstream regulatory element antagonist modulator) [8]. In 2000, An et al. used the yeast two-hybrid system to identify three proteins of 216-, 252-, and 256-amino acid length that interacted with K_V_4 channels, and named them KChIP1, KChIP2, and KChIP3. Interestingly, they found that KChIP3 was translated from the same DNA sequence as the previously reported calsenilin and DREAM [9]. These three landmark papers marked the beginning of research into the functions of KChIPs. Academic research has uncovered the functions of KChIPs, which revolve around these three aspects. First, KChIPs can interact directly with intracellular proteins to regulate various cellular functions through non-transcriptional mechanisms. For example, in neurons, KChIPs interact with PS to regulate the activity of γ-secretase [7,10]. In platelets, KChIP3 exerts a procoagulant effect by directly binding to and activating phosphatidylinositol 3-kinase-Iβ [11]. Second, KChIPs can also regulate physiological and pathological responses at the transcriptional level. In addition to KChIP3, other KChIPs also have transcriptional activity to modulate the transcription of genes regarding circadian rhythm, pain, memory, inflammatory response, immune response, and hormone secretion. Last but not least, KChIPs regulate the subcellular localization and gating properties of fast-inactivating K_V_4 channels mainly in neurons and cardiomyocytes. In addition, KChIPs are involved in the regulation of other important ion channels in the heart, including K_V_1.5, Ca_V_1.2, and Na_V_1.5. Therefore, KChIPs play a pivotal role in controlling the electrophysiological function of cardiomyocytes. Among these, the functions of KChIPs as transcriptional regulators and auxiliary subunits of ion channels have been studied the most, which is the focus of this review (Figure 1).

KChIPs have been implicated in the pathogenesis of several diseases, including cardiac arrhythmias, cardiac hypertrophy, neurodegenerative diseases, and epilepsy. As research deepens and the role of KChIPs in physiological and pathological conditions become better understood through the use of some transgenic or gene knockout animal models, the evidence for the association of KChIPs with these diseases is increasing. Small molecules acting on KChIPs have been shown to enhance or inhibit the activity of KChIPs [12], which is instructive for the development of drugs targeting KChIPs to treat related diseases. However, a growing number of recent studies have shown the diversity of functions of KChIPs. For example, KChIP3 knockout rats and mice appear to have different pain hypersensitivity responses to drug stimulation. The small molecule NS5806 has different effects in different species, even in different parts of the same species. The emergence of these questions indicates the complexity of the function of KChIPs, and there are still many questions in this field that researchers urgently need to answer. In this review, we summarized the function of KChIPs and their multiple roles in disease progression. Meanwhile, we updated the small-molecule drugs targeting KChIPs to provide new guidance for future basic and translational research in this field.

**Table 1 cells-12-01894-t001:** The expression and post-translational modification of KChIPs.

	KChIP1	KChIP2	KChIP3	KChIP4
Aliases	NA	NA	DREAM [8]; Calsenilin [7]	CALP [10]
Tissue distribution	Brain [1]; Heart [13]; Intestine [4,5]; Stomach [6]; Pancreas [14]	Brain [15]; Heart [1]; Lung [2]; Immune System [16]	Brain [1]; Heart [13,17]; Lung [2]; Immune System [16]; Nasal Mucosa [18]	Brain [1]; Kidney [3]
Post-translational modification	Myristoylation [19]	Palmitoylation [20]	Palmitoylation [20] Phosphorylation [21] Sumoylation [22]	Palmitoylation [20]
Interacted K_V_ subtypes	K_V_4.1 [23,24] K_V_4.2 [9,24,25,26] K_V_4.3 [9,23,25,26]	K_V_4.1 [27] K_V_4.2 [9,27] K_V_4.3 [27]	K_V_4.1 [28] K_V_4.2 [9,29,30,31] K_V_4.3 [32,33,34]	K_V_4.2 [30] K_V_4.3 [25,35]

NA, not available.

## 2. The Molecular Properties of KChIPs

### 2.1. The Structure of KChIPs

KChIPs are bipolar proteins of approximately 217–270 amino acids, with a C-terminal core domain and a variable N-terminal domain containing multiple post-translational modification sites (Figure 2). The C-terminal domain is highly conserved and contains a region that interacts with potassium voltage-gated channel subfamily D member (K_V_4) and four EF-hand motifs [36,37]. Of the four EF-hands, EF-3 and EF-4 can bind Ca^2+^, EF-2 can bind Mg^2+^, and EF-1 is degenerate [36,38]. KChIPs undergo conformational modifications when bound to divalent cations [39]. There are several post-translational modification sites at the N-terminus of KChIPs. For example, KChIP1 has an N-terminal myristoylation motif [19]. KChIP2 can undergo dynamic palmitoylation (Cys45 and Cys46) [20,40], namely palmitoylation and depalmitoylation. KChIP3 can undergo sumoylation (K26 and K90) [22], palmitoylation (Cys45 and Cys46) [20], and phosphorylation (Ser63 and Ser95) [21,41]. These diverse post-translational modification sites and patterns in the N-terminus determine the subcellular localization of KChIPs. Myristoylation of KChIP1, palmitoylation of KChIP2, and palmitoylation and phosphorylation (Ser95) of KChIP3 all enhance KChIPs’ localization to the cell membrane. In contrast, depalmitoylation of KChIP2 and sumoylation of KChIP3 increase their nuclear distribution. Furthermore, phosphorylation of KChIP3 at Ser63 inhibits its cleavage by caspase-3 [41].

KChIP4.4 (KChIP4a) [25] and KChIP3x (KChIP3b) [30], which possess the functional K_V_4 channel inhibitory domain (KID), an N-terminal membrane-spanning segment, are able to exert the inhibitory effect on K_V_4 channels. The KID contains an ER retention motif consisting of six hydrophobic and aliphatic residues 12–17, which interferes with K_V_4 surface expression. Residues 19–21 (VKL motif), adjacent to the ER retention motif, enhance K_V_4 inactivation and keep it in the closed state, thereby inhibiting channel current [35,37,42].

### 2.2. Regulation of KChIPs Expression

#### 2.2.1. Transcriptional Level

Some physiological functions or pathological conditions are underpinned by changes in the expression levels of KChIPs. Unfortunately, mechanistic studies of the expression regulation of KChIPs are limited and have mainly focused on KChIP2. A number of signaling pathways have been identified that affect the transcription of KChIP2, including NF-κB [43], CaMKII [44], NFAT [45], MAPK [46,47,48], and Notch [49]. Our previous work showed that NF-κB can directly bind to the promoter region of the *Kcnip2* gene and repress its transcription [43]. Several factors that downregulate KChIP2 expression, including ligands of TGF-β receptors [50], α-adrenergic receptors [51], β-adrenergic receptors [52], and C-reactive protein [53], are partially dependent on the activation of the NF-κB pathway. Additionally, Ca^2+^ reduces the mRNA expression of KChIP2 through the Ca^2+^/calmodulin-dependent protein kinase II (CaMKII) [44] and the calcineurin/NFAT signaling pathway [45]. Under physiological conditions, the inhibition of MEK1 increases, whereas the activation of MEK1 decreases KChIP2 mRNA level. In phenylephrine-induced hypertrophic cardiomyocytes, inhibition of JNK1 rescues the downregulation of KChIP2. These findings suggest that mitogen-activated protein kinase (MAPK) pathways are also involved in the regulation of KChIP2 expression [30]. Notch signaling inhibits the expression of KChIP2 expression and thereby contributes to the electrophysiological differences between neonatal and adult cardiomyocytes [49]. Several transcription factors have also been reported to bind directly to the promoter region of *KCNIP2* to regulate its expression. For example, cyclic AMP response element binding protein (CREB) [54], Krüppel-like factor-4 [55], and Krüppel-like factor-5 [56] are able to promote *KCNIP2* transcription (Figure 3).

#### 2.2.2. Protein Level

It is worth noting that as an integral component of the K_V_4 channel complex, the protein level of KChIPs is tightly coupled to K_V_4 and dipeptidyl-peptidase-related proteins (DPPs). On the one hand, deletion of K_V_4.2 significantly reduces KChIP1, KChIP2, and KChIP3 expression in the mouse brain. In particular, deletion of K_V_4.2 results in reduced expression of KChIP2 and KChIP3 in the hippocampus, KChIP2 in the striatum, and KChIP1 and KChIP3 in the cerebellum [57]. Furthermore, KChIP2, KChIP3, and KChIP4 protein expression levels in cortical pyramidal neurons were extremely low in mice with a targeted deletion of either K_V_4.2 or K_V_4.3 [15]. Nerbonne et al. showed that loss of K_V_4.2 in cortical pyramidal neurons resulted in targeted degradation of KChIP3 protein [58]. On the other hand, DPPs also affect the expression of KChIPs. Downregulation of DPP6 reduces K_V_4.2 and KChIPs in CA1 hippocampal neurons [59]. Similarly, knockdown of DPP10 in the dorsal root ganglion (DRG) neurons resulted in downregulation of KChIP1 and KChIP2 [60].

In addition, members of the KChIP subfamily can have an effect on the expression of other KChIPs. For example, when KChIP3 was deficient in the cortex, the expression of other KChIPs were increased to compensate for KChIP3 deficiency [15,61]. Interestingly, there is evidence that KChIP3 can be a negative regulator of its own expression [61].

## 3. Biological Function of KChIPs

### 3.1. KChIPs Are Auxiliary Subunits of K_V_4 Channels

#### 3.1.1. The Interaction of KChIPs with K_V_4 Channels

K_V_4 channels are members of the K_V_ channel superfamily. In mammals, the K_V_4-family consists of four members: K_V_4.1, K_V_4.2, and two splice variants of K_V_4.3. All K_V_4 channels share a functional core that is assembled as a tetramer of pore-forming α-subunits around a central pore [12]. The K_V_4 α-subunit contains an N-terminal cytoplasmic domain with an N-terminal hydrophobic segment, the K_V_ channel assembly domain (T1 domain) under the tetrameric channel pore domains, a transmembrane domain with six transmembrane helices S1–S6, and the C-terminal cytoplasmic domain [62]. The K_V_4 channels are highly expressed in the brain, heart, and smooth muscle cells. The neuronal K_V_4 channels underlie the transient A-type current (*I*_A_), sustaining the homeostatic excitability of neurons [32]. In cardiomyocytes, K_V_4 channels control the early repolarization phase of the action potential by mediating the transient outward current (*I*_to_) [63]. In gastrointestinal smooth muscle cells, K_V_4 currents are involved in shaping the slow-wave activity and mechanical responses [4].

However, K_V_4 channels can not carry out normal physiological processes on their own. Two auxiliary subunits, KChIPs and DPPs, are essential for the physiological function of K_V_4 channels. Mechanistically, KChIP1-3 can bind to the cytoplasmic domains of K_V_4 α-subunits, thereby increasing total K_V_4 current, slowing channel inactivation, and accelerating recovery from inactivation. An et al. also found that KChIP4a, which contains the KID, can eliminate the fast inactivation of the K_V_4 current. KChIPs are anchored laterally to the T1 domains of K_V_4 and clamp two adjacent K_V_4 N-terminals in a 4:4 ratio [64,65]. Recently, Kise et al. reported the cryo-electron microscopy structures of K_V_4.2–KChIP1 octamer complex. They suggested that the structure of the K_V_4.2–KChIP1 octamer complex has dimensions of around 105 Å × 105 Å × 100 Å [62]. The X-ray crystal structures of the KChIP1-K_V_4 complex have revealed some of the interaction sites that mediate the regulatory effects of KChIPs. Wang et al. reported two key sites for KChIP-K_V_4 interaction by resolving the crystal structures of the KChIP1-K_V_4 complex formed by the N-terminus of human K_V_4.3 (residues 6–145) and human KChIP1 (residues 38–217). At the first site, the hydrophobic pocket formed by the H10 helix of KChIP1 interacts with the N-terminal hydrophobic segment of K_V_4.3, which represents the main binding mode of KChIP1 with K_V_4. The N-terminal hydrophobic peptide of K_V_4.3 is sequestered in an elongated hydrophobic groove on the surface of KChIP1 and replaces the H10 helix of KChIP1. In addition, KChIP1 can interact with the neighboring K_V_4.3 N-terminals through hydrophobic interactions and hydrogen bonds. The second site is mainly formed by the interaction between the KChIP1 H2 helix and the KChIP-specific docking loop of the neighboring T1 domain through the hydrophobic interactions and salt bridges, stabilizing the K_V_4.3 tetramerization. In the second site, the conserved Phe73 of K_V_4 fits tightly into a hydrophobic cavity that is formed by the residues Leu39, Leu42, Leu53, Tyr57, and Phe108 in KChIP1. Meanwhile, residues Lys50, Arg51, and Lys61 of KChIP1 form salt bridges with Glu77, Asp78, and Glu70 of K_V_4.3, respectively [65]. The first site is responsible for K_V_4 inactivation and the second site for stabilizing the tetramer of K_V_4. Based on crystallographic studies by Wang et al., Cattee et al. defined the third site of K_V_4.3-KChIP1 interaction using all-atom molecular dynamics simulations. Residues R51, R58, and E63 on the KChIP1 H2 helix, which is also involved in the formation of the second site, interact with residues D39 and R60 on the K_V_4.3 T1N linker to form the third site, providing further stability to the K_V_4.3 tetrameric intracellular domain [65,66]. In addition, KChIP1 is able to capture the C-terminal cytoplasmic S6 helix on K_V_4.2 through the hydrophobic interactions [62].

#### 3.1.2. KChIPs Modulate the Gating Properties of K_V_4 Channels

K_V_4 are the rapidly inactivating (A-type) K_V_ potassium channels that generate currents at subthreshold membrane potentials. They are characterized by fast activation, fast inactivation, and fast recovery from inactivation. The inactivation of K_V_4 channels is classified into two types: open-state inactivation and closed-state inactivation. Closed-state inactivation is the main type, indicating that K_V_4 channels can be inactivated directly from the closed state [67]. Binding of KChIPs to the N-terminus of K_V_4 modulates the gating properties of K_V_4 channels. Specifically, KChIP1-3 augment K_V_4 currents through the following electrophysiological effects: shifting the activation midpoint of voltage activation to more negative potentials, slower inactivation, and acceleration of recovery from inactivation [9]. When co-expressed with KChIPs, the activation time of K_V_4 was slightly prolonged compared to K_V_4 alone, while the midpoint for K_V_4 of voltage activation significantly shifted to more hyperpolarized potentials [9]. In contrast, the modulation of K_V_4 gating by KChIPs is mainly manifested in the inactivation kinetics. Heterologous co-expression of K_V_4 and KChIPs significantly prolongs the inactivation time of K_V_4 channels. To be specific, KChIPs eliminate open-state inactivation and accelerate closed-state inactivation of K_V_4 channels [9,27,62]. However, it is still unclear about the molecular mechanism by which KChIPs control K_V_4 inactivation. The EF-hands were reportedly involved in the regulation of K_V_4.3 inactivation by sensing intracellular Ca^2+^ levels [68]. Recently, breakthroughs have been made in the structural basis of KChIPs that regulate the inactivation kinetics of K_V_4 channels. Kise et al. reported that KChIP1 is able to capture and sequester both the N-terminal hydrophobic segment and the C-terminus of K_V_4.2 channels. KChIP1, on the one hand, binds the C-terminal intracellular S6 helix to stabilize the S6 conformation. It also binds the N-terminal hydrophobic segment and two T1 domains from the neighboring subunit of K_V_4.2. Together, these KChIP1-mediated structural features prevent open-state inactivation and accelerate closed-state inactivation of K_V_4.2 [62]. By truncating the N-terminal or C-terminal helix of K_V_4.2, respectively, Ye et al. demonstrated that the interactions of KChIP2 with the K_V_4.2 N-terminal helix play a more prominent role in modulating channel inactivation [69]. Moreover, KChIPs accelerate the rate of recovery of K_V_4 channels from inactivation in a Ca^2+^-independent manner [68].

It is interesting to note that a specific KChIP isoform, KChIP4a, has been reported to delay K_V_4.3 channel activation, abolish rapid inactivation, and impede channel closure after opening. Therefore, co-expression of KChIP4a with K_V_4 α-subunits converts the A-type K_V_4 current to a slowly inactivating delayed rectifier-type potassium current [25]. The similar suppressive effect on K_V_4 currents was also found for KChIP3x (KChIP3b) in subsequent research [30].

#### 3.1.3. KChIPs Modulate the Trafficking of K_V_4 Channels

K_V_4 trafficking to the cell surface are vital for the maintenance of current density [9,27]. The efficient trafficking of K_V_4 to the cell surface depends on KChIP binding to its N-terminal domain [27]. Shibata et al. postulated that in the absence of KChIP binding, the hydrophobic domains of multiple K_V_4.2 subunits will oligomerize, resulting in aggregation, misfolding, and consequently retention in the ER for degradation. Interaction of KChIPs with K_V_4.2 leads to changes in the overall molecular properties of K_V_4.2 by concealing the N-terminal hydrophobic domain. These changes include an increased protein stability, phosphorylation, detergent solubility, and cell surface localization [70]. Further research has shown that in the absence of KChIP1, the K_V_4 protein can be released from the ER but is then trapped in the Golgi complex. When K_V_4 channels and KChIP1 are co-expressed in heterologous expression systems, their translocation to the plasma membrane is facilitated [27,28]. In addition, KChIP2 and KChIP3 have also been shown to increase K_V_4.2 trafficking from the ER and Golgi complex to the plasma membrane [71]. In detail, K_V_4.2 is targeted by KChIP2 to cholesterol-rich lipid rafts in the cell membrane [72]. By contrast, KChIP4a, which contains the KID motif, does not have these effects on K_V_4.2. Meanwhile, KChIP4a suppresses K_V_4 trafficking by forming a ternary plasma membrane complex with K_V_4.2 and other KChIPs [70]. Therefore, the role of KChIPs in K_V_4.2 trafficking may also contribute to its regulation on K_V_4 current (Figure 4).

#### 3.1.4. KChIP Ligands Affect KChIP Regulation on K_V_4

Since KChIPs are essential for the control of electrophysiological processes in cells, inhibiting or enhancing their effect on K_V_4 channels is a promising avenue for pharmacological research in the treatment of K_V_4-mediated channelopathies. Several small molecules that bind to KChIPs have been shown to modulate K_V_4 currents (Table 2). For example, the binding of arachidonic acid to the hydrophobic C-terminus of KChIP1 accelerates K_V_4 inactivation and decreases current amplitude [26]. CL-888, a diaryl–urea compound, can bind to KChIP1 [73] and KChIP3 [74] to counteract their regulatory effects on K_V_4, reducing peak current amplitude and accelerating inactivation kinetics. IQM-PC330 and IQM-PC332, the derivatives of CL-888 modified by Lopez-Hurtado et al., have a more refined blocking effect on K_V_4 currents. IQM-PC330 and IQM-PC332 inhibit K_V_4.3 channels not only by reducing peak current amplitude and accelerating their inactivation but also by delaying their recovery from inactivation. Mechanistically, they act as KChIP3 ligands to reverse the regulatory function of KChIP3 on K_V_4.3 channel gating properties. It is interesting to note that the hydrophobic combination of IQM-PC332 and KChIP3 accelerates the activation kinetics of K_V_4.3 current at low concentrations (0.01 to 0.1 M) and reverses its effect on channel gating at higher concentrations. However, inactivation recovery kinetics were significantly reduced at all concentrations [75]. They also identified IQM-266 as a novel KChIP3 ligand with similar inhibitory properties to IQM-PC330 and IQM-PC332 [76]. Recently, IQM-266 was reported to bind KChIP2 and increase K_V_4.3/KChIP2 currents [77].

The sulfonylurea compound NS5806, a ligand of KChIP3, has been demonstrated to activate K_V_4.3 channels in neurons. NS5806 delays the inactivation of *I*_A_ and slightly reduces the maximum peak current [78]. This effect is based on the Ca^2+^-dependent binding of NS5806 to the hydrophobic site at the C-terminus of KChIP3, which facilitated the binding affinity between KChIP3 and K_V_4.3 as well as reducing their dissociation rate [34]. In cardiomyocytes, however, the control of K_V_4 currents by NS5806 is controversial. In canine cardiomyocytes, NS5806 functions as an *I*_to_ activator to increase the amplitude of the K_V_4.3/KChIP2 peak current and significantly slow the current decay [79], while in mouse ventricular myocytes NS5806 has the opposite effect, as shown by a decrease in the amplitude of native *I*_to_ and significant acceleration of current inactivation [80]. Furthermore, in rabbit ventricular myocytes, NS5806 dramatically raised *I*_to_ amplitude, while in atrial myocytes, the *I*_to_ amplitude was repressed [81]. These conflicting results suggest that more than one site may be involved in the interaction between NS5806 and K_V_4 channel complexes, and that NS5806 may act as both an agonist and an antagonist of *I*_to_ on the same channel complexes. The exact mechanism by which NS5806 elicits different responses in different species and cell types remains unclear. Indeed, studies comparing the effects of reported small molecule drugs targeting KChIPs in different species are still lacking. It is therefore necessary to test and compare these small molecules in more animal models. More interestingly, repaglinide, a commonly used antidiabetic drug, can bind directly to KChIP3 and exert an inhibitory effect on K_V_4 channels [74]. In fact, accumulated research has shown that repaglinide and glibenclamide can competitively bind to the SUR subunit of cardiovascular and neurological K_ATP_ (ATP-sensitive potassium) channels to inhibit *I*_KATP_ [82,83]. However, the magnitude and kinetics of K_V_4 currents were not affected by glibenclamide [74]. Mechanistically, repaglinide can bind to the Ca^2+^-dependent EF-hand protein of the neuronal calcium sensor family to antagonize its biological function [84]. It is regrettable that the effects of repaglinide against other KChIPs in other organs have not been reported. Given that repaglinide is already a mature drug that can improve type 2 diabetes and cardiovascular events, further investigation of the effect of repaglinide on K_V_4 currents in different tissue will be a potential way to improve KChIPs-related neurological and cardiovascular diseases. Altogether, the discovery of these KChIP ligands has improved our knowledge of the interaction between K_V_4 and KChIPs and has paved the way for future pharmaceutical development for the treatment of neurodegenerative and cardiovascular diseases involving K_V_4/KChIPs.

**Table 2 cells-12-01894-t002:** Small-molecule compounds that interact with KChIPs and their role in K_V_4 currents.

Compounds	KChIP1	KChIP2	KChIP3	KChIP4
Arachidonic acid	Accelerate K_V_4 inactivation; Reduce current amplitude [26]	NA	NA	NA
CL-888	Reduce peak current amplitude; Accelerate inactivation kinetics [73]	NA	Reduce peak current amplitude; Accelerate inactivation kinetics [74]	NA
IQM-PC330 IQM-PC332	NA	NA	Reduce peak current amplitude; Accelerate inactivation kinetics; Delay recovery from inactivation [75]	NA
IQM-266	NA	Increase current [77]	Reduce peak current amplitude; Accelerate inactivation kinetics; Delay recovery from inactivation [76]	NA
NS5806	NA	Play opposite roles in different species, even in different tissues from one species [79,80,81]	Activate neural K_V_4.3; Delay inactivation kinetics; Reduce the maximum peak current slightly [78]	NA
Repaglinide	NA	NA	Inhibit K_V_4 channels [74]	NA

NA, not available.

### 3.2. Role of KChIPs in Regulating Other Ion Channels

#### 3.2.1. K_V_1.5

K_V_1.5 is another vital potassium channel characterized by rapid activation and rapid inactivation that is highly expressed in both the brain and heart. In the human heart, K_V_1.5 is expressed abundantly in atrial myocytes and mediates the ultra-rapid delayed rectifier current (*I*_Kr_). In the adult mouse heart, K_V_1.5, which encodes the slow delayed rectifier K^+^ current (*I*_Ks_), is involved in the repolarization of ventricular myocytes [85]. Several studies have demonstrated the contribution of KChIPs in the modulation of K_V_1.5 channels. In contrast to their augmenting effect on K_V_4 currents, KChIPs negatively regulate K_V_1.5-encoded K^+^ currents. In transiently transfected HEK293 cells, KChIP1 and KChIP2 attenuate K_V_1.5 currents by inhibiting the trafficking of K_V_1.5 channels to the cell surface [86]. Moreover, in the ventricles of *Kcnip2*^−/−^ mice, K_V_1.5 mRNA levels were significantly increased and *I*_Ks_ were upregulated [87]. However, the structural basis of KChIP1 and KChIP2 negative regulation of cardiomyocyte K_V_1.5 channels in vivo remains unclear and needs further investigation.

#### 3.2.2. Ca_V_1.2

The high-voltage-activated L-type Ca^2+^ channel, Ca_V_1.2, located on the t-tubule sarcolemma, is the major calcium channel type in cardiomyocytes. Ca_V_1.2 channels are macromolecular complexes that are composed of α_1_, α_2_δ, and β subunits [88]. The cardiac L-type Ca^2+^ current (*I*_Ca,L_) mediated by Ca_V_1.2 is essential for cardiomyocyte depolarization and contraction. KChIPs have a dual effect on the regulation of the cardiac L-type Ca^2+^ current (*I*_Ca,L_). On one hand, KChIP2 regulates the *I*_Ca,L_ through direct interaction with the intracellular N-terminus of the Ca_V_1.2 α_1C_ subunit. Without raising Ca_V_1.2 protein expression or trafficking to the plasma membrane, KChIP2 increases *I*_Ca,L_ current density by impeding the N-terminal inhibitory module [89]. On the other hand, KChIP2 and KChIP3 bind and repress the transcription of the *Cacnb2* [90] and *Cacna1c* [17] genes, respectively, which encode the corresponding β_2_-subunit and α_1C_-subunit of Ca_V_1.2 channels.

#### 3.2.3. Na_V_1.5

Na_V_1.5, which is encoded by *SCN5A*, is highly expressed in the heart. By mediating the rapid influx of Na^+^, Na_V_1.5 dominates the rapid depolarization phase of action potential. Previous studies suggest that K_V_4.3 and Na_V_1.5 work coordinately and can regulate each other [91]. Deschênes et al. found that co-expression of KChIP2 with Na_V_1.5 increased Na_V_1.5 current density but had no effect on Na^+^ current gating properties, whereas silencing of *Kcnip2* resulted in a significant decrease in *Scn5a* mRNA level and complete inhibition of voltage-dependent Na^+^ currents [92]. However, this evidence is based on heterologous expression systems. The electrophysiological significance of the cross-regulation of KChIP2, Na_V_1.5, and K_V_4.3 in cardiomyocytes requires further study.

### 3.3. KChIPs Are Ca^2+^-Dependent Transcriptional Factors

KChIP3 is the first KChIP identified to have transcriptional activity, as it can bind to the downstream regulatory element (DRE) downstream of the TATA box in the human *PDYN* gene promoter [8]. Subsequent studies have demonstrated that all the KChIPs have Ca^2+^-dependent DRE binding affinities that block gene transcription in the form of homotetramer or heterotetramer. The binding of KchIP3 to DRE sequences is regulated by divalent cations [8]. Tetramer formation is required for KchIP3 in binding to DRE. Mg^2+^ and Ca^2+^ are the most important factors affecting the ability of KchIP3 to bind to DNA. The combination of Mg^2+^ and KchIP3 stabilizes the tertiary structure of the protein and promotes DNA binding [93]. In contrast, as intracellular Ca^2+^ levels increase, the Ca^2+^-bound KchIP3 switches to a denser dimer structure [94], preventing KchIP3-DRE interaction. Furthermore, direct interaction of nucleoprotein C-terminal binding protein [95] and αCREM [96] with KChIP3 enhances and inhibits their transcriptional repressive functions, respectively. In the nucleus, in addition to direct binding to DRE in target genes, KChIP3 also forms transcriptional regulatory complexes with nuclear proteins such as cAMP response element-binding protein [97], thyroid transcription factor 1 [98], and nuclear receptors [99]. Genes regulated by KChIP3 include but are not limited to the following: *Pdyn*, *Fos* [100], *Hrk* [101], *Bdnf* [102], *Ifng*, *Il2* and *Il4* [103], *Klf9* [16], *Npas4*, *Nr4a1*, *Mef2c*, *Junb* [61], *Tnfaip3* [104], *Tg* [98], *Ttf2*/*Foxe1*, *Pax8* [105], *Aanat*, *Fra-2*, *Crem* [106], *GnRH* [107], *Gfap* [108], *Ncx3* [109], *Cacna1c* [17], *Mid1* [110], *Cant1* [111], *PAX6*, *NRG1* [112], and *GCM1* [113] (Figure 5).

Through the transcriptional regulation of the above-mentioned genes, KChIP3 has been shown to be multifunctional and is critically involved in the regulation of physiological and pathological functions of neurons, endocrine cells, immune cells, endothelial cells, and hematopoietic cells. For example, in the pineal gland, KChIP1-4 are involved in the regulation of rhythmically expressed genes engaged in circadian rhythms. By binding to the DRE sites of arylalkylamine N-acetyltransferase (*Aanat*), inducible cAMP early repressor (*Crem*), and Fos-related antigen-2 (*Fra-2*), KChIP1-4 are able to repress the basal and induced transcription of these circadian-rhythm-related genes [106]. In the pancreas, KChIP3 has been detected in islet α- and β-cells. KChIP3 represses the transcription of the *Pdyn* gene in a Ca^2+^-dependent manner and thereby affects glucagon release [114]. In the thyroid, KChIP3 inhibits the expression of thyroglobulin by binding directly to the DRE of the thyroglobulin gene and blocking thyroid-specific transcription factors such as TTF-1, TTF-2, and Pax8 [98,105]. KChIP3, as well as KChIP2, are expressed in T- and B cells to regulate immunological responses. In T lymphocytes, KChIP2 and KChIP3 act as transcriptional repressors to inhibit the expression of IFN-γ, IL-2, and IL-4 [103]. By inhibiting the transcription of the proliferation-related gene *Klf9* and the protein-translation-related gene *Eif4g3* in B cells, KChIP2 and KChIP3 govern B-cell proliferation and IgM and IgG protein synthesis [16]. In lung endothelial cells, neutrophils, and macrophages, KChIP3 can promote the NF-κB-pathway-mediated inflammatory response by suppressing the expression of the *TNFAIP3* gene, which encodes the anti-inflammatory deubiquitinase A20 [104,115].

## 4. KChIPs and Diseases

### 4.1. Neurological Diseases

#### 4.1.1. Epilepsy

The International League Against Epilepsy (ILAE) classification of epilepsy types includes focal epilepsy, generalized epilepsy, combined generalized and focal epilepsy, and unknown epilepsy [116]. All of these seizure types share the pathophysiological characteristic of increased neuronal excitability and synchronicity. Among them, temporal lobe epilepsy is the most common focal epilepsy that can further be subdivided into mesial temporal lobe epilepsy and lateral or neocortical temporal lobe epilepsy [117]. Adults with intractable epilepsy are most commonly affected by epilepsy of the mesial temporal lobe, which is the chronic and pharmacoresistant form of epilepsy. In this type of epilepsy, seizures originate from the hippocampus, entorhinal cortex, amygdala, and parahippocampal gyrus [118]. Abundantly expressed in the hippocampus, KChIPs control the frequency of slow repetitive spike firing and attenuate action potential backpropagation by modulating the properties of K_V_4 channels. Thus, KChIPs play a key role in maintaining neuronal excitability [119]. To date, several studies have implicated the downregulation of KChIPs in temporal lobe epilepsy. For example, the hippocampus of patients with medically refractory temporal lobe epilepsy had significantly reduced KChIP3 immunoreactivity compared to normal brains [120]. Consistently, downregulation of KChIP expression in specific hippocampal subfields (CA1 and CA3, but not CA2) was observed in rodent models of temporal lobe epilepsy that progressed to status epilepticus, i.e., very prolonged seizures. For example, in the hippocampus of the pilocarpine-induced rat epilepsy models, loss of KChIP1 immunoreactivity in interneurons and reduction of KChIP2 in the stratum radiatum of the CA1 region were observed [121]. In the kainic-acid-induced mouse epilepsy model, KChIP3 expression was reduced in the cortical area and CA3 region of the hippocampus in status epilepticus [120]. The use of KChIP-deficient transgenic mice provided further evidence for the role of KChIPs in the pathogenesis of epilepsy. In *Kcnip2*^−/−^ mice, the excitability of hippocampal neurons was enhanced and the susceptibility to epilepsy induced by kindling was increased. In the hippocampal pyramidal neurons from *Kcnip2*^−/−^ mice, *I*_A_ showed a reduced amplitude and shift in *V*_½_ for steady-state inactivation to hyperpolarized potentials [122]. Nevertheless, the role of KChIPs in the pathogenesis of epilepsy remains to be further elucidated using neuron-specific *Kcnip* overexpression or knockout animal models.

#### 4.1.2. Pain

A key factor in the development and maintenance of neuropathic pain is neuronal excitability. The physiological pain circuit can be briefly summarized as follows: physicochemical signals from noxious stimuli transduces through ion channels and purinergic channels to evoke action potentials. These action potentials are amplified by Na^+^ channels to produce pain. The electrical signals are carried by unmyelinated C-fibres and thinly myelinated Aδ-fibres to the DRG in the body and the trigeminal ganglion (TG) in the face, where their cell bodies are located and project to the dorsal horn of the spinal cord and the medulla oblongata, respectively. After integration and processing, the pain input is transmitted via several ascending tracts to various projection sites in the brain. To process the sensory and discriminative aspects of pain, the lateral spinothalamic tract projects to the lateral thalamus. Medial projections of the spinothalamic and parabrachial tracts to the medial thalamus and limbic structures mediate the emotional and aversive components of pain. In pathological conditions such as inflammation, neuropathy, and diabetes, physiological pain is converted into pathological pain, which manifests as increased sensitivity to painful stimuli (hyperalgesia) [123].

As proposed by Costigan et al.: “No DREAM, no pain” [124], indicating that KChIP3 is critically involved in the regulation of pain. KChIP3 is expressed in the neurons of the ventral and dorsal horns of the spinal cord and the TG of mice [22,100]. Using *Kcnip3*^−/−^ mice, Cheng et al. proposed that loss of KChIP3 attenuates pain responses. *Kcnip3*^−/−^ mice showed significantly reduced pain behaviors in models of visceral pain induced by MgSO_4_ and acetic acid, chemical pain induced by formalin, inflammatory pain induced by capsaicin and carrageenan, and neuropathic pain induced by cuff implantation around the sciatic nerve [100]. This result was further substantiated by Rivera et al. using daDREAM (dominant active DREAM)-transgenic mice. daDREAM is a Ca^2+^-insensitive double mutant (EF-hand, leucine-charged residue-rich domains) KChIP3 that actively represses KChIP3 target genes and prevents the Ca^2+^-dependent derepression function of KChIP3. In contrast to *Kcnip3*^−/−^ mice, which showed a basal state of analgesia, daDREAM transgenic mice displayed a state of basal hyperalgesia. However, the daDREAM transgenic mice showed impaired response to inflammatory pain induced by Complete Freund’s adjuvant (CFA) [102]. In the above studies, the modulation of the basal threshold of pain by KChIP3 was attributed to the inhibition of the expression of PDYN, which can be cleaved into dynorphin in the spinal cord and acts as an endogenous ligand of κ-opioid receptors to exert analgesic effects [125]. Meanwhile, the regulation of spinal sensitization by KChIP3 was considered to be dependent on BDNF which is a well-established regulator of synaptic plasticity and plays a modulatory role in spinal nociceptive processing. BDNF in the physiological dose range facilitates nociception, whereas high doses produce analgesia or hypoalgesia [126]. Both *Pdyn* and *Bdnf* promoter regions contain the DRE sequence [8,61], the transcription of which could be inhibited by KChIP3. Notably, in daDREAM transgenic mice, orofacial injection of formalin (acute trigeminal nerve stimulation) induced hyperalgesia with a concomitant reduction in BDNF [127]. This finding was contrary to that reported by Rivera above, which may be because BDNF plays different and specific roles in the control of nociception in the trigeminal ganglion and in the spinal cord. Furthermore, in trigeminal neurons, KChIP3 specifically downregulates *Mgll* (encodes monoglyceride lipase) and *Ctsl* (encodes cathepsin L), which are associated with nociception [127].

Most of the conclusions drawn from the mouse model have been confirmed in the rat model. For example, upregulation of KChIP3 in the nuclear compartment was observed in a formalin-injection-induced acute pain model in rats [128]. Minocycline administration attenuated tactile allodynia and chemical hyperalgesia in diabetic rats, accompanied by downregulation of KChIP3 protein in the spinal cord [129]. However, the “no DREAM, no pain” hypothesis does not seem to apply to *Kcnip3*^−/−^ rats. Guo et al. recently found that global knockout of KChIP3 using CRISPR/Cas9 technology increased pain sensitivity in rats to acute and chronic inflammatory pain induced by formalin and CFA [130,131], which is in contrast to what was found in studies of *Kcnip3*^−/−^ mice by Cheng et al. [100]. This is a big wake-up call that KChIP3, in addition to repressing the expression of genes involved in pain processing, may have other functions in pain transmission and processing in rats. But the specificity of KChIP3 in nociception and pain processing at different levels could not be demonstrated in transgenic animals with global KChIP3 knockout. Indeed, KChIP3 is highly expressed in rat DRG neurons [132], whereas previously reported KChIP3 protein in mouse DRG could not be detected by Western blot [100]. Some reports suggest that the mechanism of KChIP3 in pain modulation in rats involves its interaction with several receptors and ion channels involved in pain-sensing and transmission in DRG neurons. These include the N-methyl-D-aspartate receptor (NMDAR) [133], transient receptor potential vanilloid 1 (TRPV1) channel [131], and K_V_4 [132]. Activation of NMDAR in the central nervous system is required for central sensitization. KChIP3 significantly inhibited the surface expression of NMDARs and NMDAR-mediated currents by directly interacting with NMDAR through its N-terminal residues 21-40 [131]. TRPV1, which is also known as the thermosensitive protein, can sense noxious heat and chemical stimuli simultaneously. In primary sensory neurons, TRPV1 acts as an integrator of the painful afferent sensation and as a key initiator of the efferent neurogenic inflammation [134]. In the rat inflammatory pain model induced by CFA, TRPV1 in nociceptive sensory neurons undergoes functional sensitization, resulting in Ca^2+^ influx. Meanwhile, upregulation of KChIP3 protein expression resulted in enhanced binding of KChIP3 to the TRPV1, inhibiting TRPV1 cell surface localization and thereby exerting an analgesic effect [131]. Furthermore, by regulating K_V_4 in DRG neurons that meditate somatodendritic *I*_SA_, KChIP3 is involved in the transmission of pain signals [132]. Notably, Ca_V_3 channels and K_V_4 channels, the important contributors to pain control, were found to form a signaling complex in rat cerebellar, hippocampal, and neocortical regions [135]. Recently, a novel KChIP3 ligand, IQM-PC332, was reported to reduce mechanical hypersensitivity in rats subjected to chronic constriction injury of the sciatic nerve. Mechanistically, IQM-PC332 binds to KChIP3 and reduces ionic currents mediated by TRPV1 channels, K_V_4.3 channels, low-voltage-activated T-type Ca^2+^ channels, and high-voltage-activated Ca^2+^ channels in DRG neurons [136]. This is a potential chemical tool and the elucidation of the structural basis of the analgesic effect of IQM-PC332 will be one of the possible steps towards a better understanding of how KChIP3 controls pain. In conclusion, KChIP3 may act at multiple levels to modulate pain. It will be necessary to use tissue-specific KChIP3 knockout models or site-specific inhibition of KChIP3 to further elucidate its complex mechanism in pain modulation (Figure 6).

#### 4.1.3. Memory Dysfunction

Learning is the process by which new information about the world is acquired, and memory is the process by which knowledge is stored. The cellular basis of learning and memory is thought to be the process by which synapses undergo bidirectional changes in synaptic strength, known as synaptic plasticity. Among the different types of synaptic plasticity, two opposite forms of synaptic plasticity, long-term potentiation (LTP) and long-term depression (LTD), have been the most studied [137]. LTP is the strengthening of synapses following repeated synaptic activity. Inhibition of LTP results in impaired learning or failure to retain memories, while LTD is the synaptic framework of weakening synaptic strengths, contributing to learning by engaging in a functional interplay with LTP [138].

Several brain regions are involved in memory, including the hippocampus, neocortex, amygdala, basal ganglia, and cerebellum. Of these, the hippocampus is essential for the spatial representation of the environment and the ability to recall specific events, or ‘episodic memory’ [139]. KChIP3 is highly expressed in the hippocampus and is also involved in the regulation of learning and memory. Alexander et al. reported that KChIP3 knockout mice exhibited enhanced memory in a contextual fear-conditioning paradigm. They also found that translocation of KChIP3 from the membrane to the nucleus was increased in mouse hippocampal neurons following the fear conditioning training paradigm [33]. In the cell nucleus, KChIP3 functions as a transcriptional repressor to regulate the expression of genes involved in memory formation, such as *Pdyn*, *Fos*, *Bdnf,* and *Npas4* [33,61]. In addition, Fontán et al. reported that loss of KChIP3 not only enhances LTP and improves learning and memory in young mice, but also improves cognition and slows age-related brain degeneration in old age [140]. Mechanistically, KChIP3 interacts with CREB, a key transcription factor involved in memory, and negatively regulates CREB-dependent transcription. This raises the threshold for CREB activation in learning and memory. Therefore, in the *Kcnip3*^−/−^ mice, the threshold for CREB phosphorylation and CREB-CBP interaction in learning and memory is lowered, and the timing of CREB phosphorylation and CBP recruitment to CREB is shortened, thereby promoting CREB-dependent transcription during learning. Consistent results were also confirmed in daDREAM-transgenic mice. Transgenic mice expressing the daDREAM showed significant impairments in learning and memory [61].

The mechanism by which KChIP3 is involved in memory regulation also involves its protein–protein interactions. NMDAR is the major ionotropic glutamate receptor in the central nervous system and plays an important role in both synaptic transmission and plasticity [141]. NMDARs are required for LTP and LTD of synaptic transmission [142]. When Ca^2+^ binds to KChIP3, the interaction between KChIP3 and PSD95 is released, allowing NMDA to function as a normal receptor. The EF-hands mutant KChIP3, which lacks Ca^2+^ binding ability, inhibits NMDAR function in mouse hippocampal CA1, impairs LTD, and thereby reduces consolidation of hippocampus-dependent contextual fear-conditioned reflexive memory [143]. In addition, K_V_4.2 channels play an important role in both synaptic plasticity and cognition, which in part contributes to the regulatory effect of KChIP3 on learning and memory [144].

#### 4.1.4. Alzheimer’s Disease

Amyloid beta-protein (Aβ) deposition is the main feature of AD, which has a toxic effect on neurons [145]. PS is the catalytic core of the high-molecular-weight enzyme complex γ-secretase, which can mediate γ-cleavage of β-amyloid precursor protein to produce Aβ [146]. KChIP3 was initially discovered as being capable of binding to the C-terminal fragment of presenilin 1 (PS1) and PS2 via the C-terminal 103 amino acids encoded by ALG3 [7]. The binding of KChIP3 to PS is dependent on the presence of Ca^2+^ [147]. Existing studies have shown that mutations of PS1 and PS2 lead to increased Aβ formation and apoptosis, thus causing early familial Alzheimer’s disease (AD) [148]. In the brains of patients with AD, the expression of KChIP3 was increased in neurons and reactive astrocytes, which is consistent with the regions with pathological changes of AD [149]. This suggests that KChIP3 may be involved in the pathogenesis of AD. Zaidi et al. subsequently showed that KChIP3 stabilizes the structure of the PS in the cerebellum and hippocampus [150]. Jo et al. reported KChIP3 as a pro-apoptotic protein that facilitates cell death induced by Aβ production [151]. Overexpression of KChIP3 increased the enzymatic activity of the presenilin-γ-secretase complex in HEK293 cells [152]. Aβ increased KChIP3 expression in cultured neurons while blocking its expression protected neuronal cells from Aβ toxicity [149]. Repaglinide has been shown to inhibit the KChIP3-PS2 interaction, suggesting a novel avenue for future AD treatment [153].

KChIP3 also regulates the intracellular Ca^2+^ signaling, the dysregulation of which has been implicated in the development of AD [154]. In AD models, increased Ca^2+^ release from ryanodine-sensitive Ca^2+^ stores have been observed in neurons [155]. Lilliehook et al. showed that overexpression of KChIP3 can enhance apoptosis by increasing ER Ca^2+^ release in glioma cells [156]. Fedrizzi et al. reported that transient co-expression of KChIP3 with PS facilitated Ca^2+^ release from the ER in neurons [157]. Mechanistically, KChIP3 binds to the DRE sequence of the isoform 3 of the Na^+^/Ca^2+^ exchanger (*Ncx3*) gene to repress its transcription, thereby regulating intracellular Ca^2+^ homeostasis [109]. In mouse hippocampal and cortical neurons, KChIP3 regulates intracellular calcium-induced Ca^2+^ release through direct protein–protein interaction with the ryanodine receptor [158]. Furthermore, KChIP3 appears to be involved in age-related brain degeneration [140], which is a well-known clinical symptom of AD. Although the current molecular understanding of the relationship between KChIP3 and AD pathogenesis is limited and insufficient, these findings clearly support KChIP3 as an appropriate target for further research into AD. Therefore, unravelling the diversity and complexity of KChIP3’s functions would pave the way for developing more effective treatments for AD.

#### 4.1.5. Other Neurodegenerative Disorders

In addition to AD, KChIP3 has been implicated in the pathogenesis of other neurodegenerative disorders. Recent evidence suggests that KChIP3 is closely linked to Huntington’s disease (HD) [159] and amyotrophic lateral sclerosis (ASL) [160]. Firstly, the levels of KChIP3 protein are significantly reduced in the hippocampus samples of HD patients. To be specific, KChIP3 interacts with activating transcription factor 6 (ATF6) to suppress the pro-survival unfolded protein response, a common feature of neurodegenerative diseases. In the mouse model of HD, inhibition of the KChIP3-ATF6 interaction delays the onset of cognitive deficits. In addition, treatment with repaglinide has been shown to delay the onset and progression of motor and cognitive decline and extend lifespan by blocking the interaction of KChIP3 and ATF6 [74,159], suggesting that KChIP3 may be a novel target for the treatment of related diseases. Second, upregulation of KChIP3 has been observed in motor neurons and astrocytes in the spinal cord and frontal cortex of ALS patients [160]. The main feature of ALS is the relentless loss of motor neurons and increased reactive astrogliosis [161]. Cebolla et al. showed that KChIP3 is able to promote astrocyte differentiation of cortical precursors via cAMP-dependent calcium signaling. The neonatal cortex of *Kcnip3*^−/−^ mice has a reduced number of astrocytes and an increased number of neurons [108]. Nevertheless, studies linking KChIP3 to these diseases have been limited. More evidence is needed to elucidate the mechanisms by which KChIP3 is involved in these neurodegenerative diseases.

### 4.2. Cardiovascular Diseases

#### 4.2.1. Arrhythmias

Cardiac arrhythmias are classified mechanistically into two categories: focal activity due to enhanced or abnormal pulse generation and reentry due to conduction disturbances [162]. Early afterdepolarizations preceding full repolarization and delayed afterdepolarizations occurring after full repolarization are the most common causes of focal arrhythmias. Prolongation of action potentials caused by *I*_Na,L_ (late Na^+^ current), the *I*_Ca,L_, or *I*_NCX_, or decreases in the repolarizing potassium currents (*I*_Kr_, *I*_Ks_, *I*_K1_) can lead to early afterdepolarizations. As mentioned above, KChIPs have been shown to regulate a wide variety of ion channels in cardiomyocytes, including those that control the depolarization and repolarization of cardiac action potentials. Thus, KChIPs are critically involved in the pathogenesis of arrhythmias, as demonstrated by some of the available evidence, either directly or indirectly. For example, KChIP2 was upregulated in aging porcine atria [163] and was significantly reduced in TLR4-activated inflammatory responses [164] and chronic NMDAR activation [165]. Atrial fibrillation could be induced in zebrafish hearts overexpressing *KCNIP1* [13]. Consistently, *Kcnip2*^−/−^ mice have an increased susceptibility to arrhythmias, manifested by a prolonged elevation in the ST segment on the electrocardiogram [166]. In fact, the ST segment coincides in time with the action potential plateau, and increasing outward current (mainly *I*_to_ and *I*_K-ATP_) or decreasing inward current (mainly *I*_Na_ and *I*_Ca_) can promote the occurrence of ST segment elevation [167]. Therefore, *Kcnip2*^−/−^ mice may have compensatory remodeling against the loss of *I*_to_. Mechanistically, KChIP2 on the one hand directly regulates the subcellular localization and gating properties of several ion channels as an auxiliary subunit in the cardiomyocyte. On the other hand, KChIP2 also controls the expression of ion channel subunits at the transcriptional level. In ventricular myocytes from *Kcnip2*^−/−^ mice, *I*_to,f_ and *I*_Na_ are abolished, *I*_Ca,L_ is downregulated, and *I*_Ks_ and *I*_to,s_ (the slow transient outward K^+^ current) are upregulated [87,92,168]. Recently, KChIP2 was reported to act as a transcriptional repressor by binding directly to the promoters of miR-34b/c, a miRNA that directly affects *SCN5A* (Na_V_1.5), *SCN1B* (Na_V_β1), and *KCND3* (K_V_4.3) gene expression. Inhibition of miR-34b/c can block the induction of arrhythmia [169]. Whether there are other potential ion channel proteins, or whether the genes encoding these ion channels are regulated by KChIP2, is a topic that deserves further investigation. Due to the multiple roles of KChIP2 in regulating ion channels in the heart, a thorough understanding of the detailed molecular mechanisms by which abnormal KChIP2 levels increase arrhythmias susceptibility is key to the development of novel therapies for the prevention and treatment of cardiac arrhythmias.

#### 4.2.2. Cardiac Remodeling

Cardiac remodeling is defined as persistent changes in cardiac structure and function in response to physiological or pathological stimuli. Pathophysiological events that cause a decrease in contractility and/or an increase in wall stress, such as ischemia/reperfusion, myocardial infarction, pressure and volume overload, hypertension, and neuroendocrine stimulation, often result in adverse cardiac remodeling [170]. KChIP2 expression is altered in many cardiovascular events, including ischemic cardiomyopathy [171], myocarditis [172], mitral valve disease [173], inflammatory cytokine-induced myocardial injury [50,174], myocardial infarction [175], and type 2 diabetes mellitus [176]. Previous studies have shown that KChIP2 expression is significantly downregulated in hypertrophic myocardium, which is associated with reduced *I*_to_ [166,177]. Some of these mechanisms have been validated in vitro models of hypertrophy. For example, overactivation of the JNK pathway was found to cause downregulation of KChIP2 in neonatal ventricular myocytes treated with phenylephrine, a robust inducer of hypertrophy [46]. In addition, our previous study demonstrated that the KChIP2 expression was decreased in phenylephrine-induced hypertrophic cardiomyocytes. Mechanistically, phenylephrine-activated NF-κB binds to the promoter region of the *Kcnip2* gene and directly represses its transcription. Overexpression of muscle-specific mitsugumin 53 upregulates KChIP2 through inhibition of NF-κB and thereby reversed phenylephrine-induced cardiomyocyte hypertrophy [43]. In addition, Jin et al. found that adenoviral overexpression of KChIP2 in vivo significantly attenuated the development of left ventricular hypertrophy in aortic-banded rats. This protective effect of KChIP2 was achieved by inhibiting MAPK signaling activity and reducing calcineurin/NFAT expression [178].

Cardiac memory is a specific form of cardiac remodeling that is manifested by the persistence of inverted T waves after the restoration of sinus rhythm. The T wave “remembers” the QRS complex from the paced or arrhythmia phase following a short alteration in the sequence of ventricular depolarization caused by pacing or arrhythmia [179]. An important electrophysiological mechanism responsible for cardiac memory is the reduction in *I*_to_ density and its significantly prolonged recovery from inactivation [180]. The decreased expression of KChIP2 during this phase is essential for the occurrence of cardiac memory. According to Ozgen et al., left ventricular pacing induces the degradation of the transcription factor CREB by initiating the production of myocardial angiotensin II and the synthesis of reactive oxygen species, which results in the downregulation of KChIP2 expression [54].

#### 4.2.3. Heart Failure

There is a growing number of evidence that KChIP2 also plays an important role in the pathogenesis of heart failure. The mRNA and protein expression of KChIP2 was significantly downregulated in the failing heart [181]. Meanwhile, autoantibodies against KChIP2 have been detected in patients with dilated cardiomyopathy. In vitro incubation of anti-KChIP2 antibody facilitates necrotic cell death in rat cardiomyocytes, suggesting that KChIP2 autoantibodies may be involved in the pathogenesis of dilated cardiomyopathy [182]. Candesartan, the angiotensin Ⅱ receptor blocker, attenuates KChIP2 downregulation in dilated cardiomyopathy and is involved in preventing severe electrical remodeling in inherited dilated cardiomyopathy [183]. Nassal et al. found that reduction of KChIP2 in guinea pig cardiomyocytes significantly increased *I*_Ca,L_ and prolonged action potentials by increasing Ca_V_1.2 protein expression [184]. They also found that KChIP2, like the neuronal KChIP isoforms, can regulate ryanodine receptor activity by interacting with PS. Loss of KChIP2 resulted in reduced ryanodine receptor activity due to a decrease in its binding affinity to PS, which disrupted calcium-induced calcium release events. This further leads to impaired contractility of cardiomyocytes and promotes the onset of heart failure [185]. Interestingly, Speerschneider et al. showed that although KChIP2 is downregulated in heart failure, the reduction of *I*_to,f_ does not promote the development of heart failure. On the contrary, reduction of *I*_to_ exerts antiarrhythmic effects in mouse heart [186]. Likewise, Grubb et al. showed that while the downregulation of repolarization currents in heart failure was exacerbated in *Kcnip2*^−/−^ mice, there was less prolongation of action potentials associated with heart failure due to compensation by upregulation of *I*_Ks_ and *I*_to,s_ [187]. In summary, the role of KChIP2 in heart failure remains controversial. Further research is needed to determine whether targeting KChIP2 has therapeutic implications in heart failure.

## 5. Concluding Remarks

KChIPs are collections of multifunctional proteins involved in a wide range of physiological processes and pathological diseases. Thanks to the contributions of many outstanding researchers in this field over the last 25 years, the understanding of their structure, regulation, biological function in health and diseases, and small-molecule drugs targeting KChIPs has been achieved. KChIPs are auxiliary subunits of K_V_4 channels and are critical for the trafficking and gating properties of K_V_4 channels. Meanwhile, KChIPs are also transcriptional regulators that repress the transcription of target genes in various systems throughout the body to maintain normal physiological functions, including the nervous, cardiovascular, respiratory, immune, and endocrine systems. To date, an increasing number of reports have partially linked dysregulation of KChIPs expression to disease. These include AD, HD, epilepsy, and memory dysfunction in the nervous system, arrhythmia, remodeling, and heart failure in the cardiovascular system. Some small-molecule drugs that regulate them have been shown to alleviate symptoms or provide therapeutic benefits in these diseases, most notably repaglinide, which has the most significant therapeutic effect in HD. In addition, many small molecule ligands targeting KChIPs have emerged in recent years, and basic research has shown good effects in activating or inhibiting KChIPs. However, our understanding of KChIPs, a class of proteins with similar functions but different characteristics, is still only the tip of the iceberg. There are still many questions that researchers need to answer. For example, the structural basis for the regulation of K_V_4 channels by KChIPs as auxiliary subunits remains elusive. The regulatory function of KChIP2 on K_V_1.5 and Na_V_1.5 found in the heterologous expression system has not yet been verified in vivo. Altered expression of KChIPs has been found in many clinical disease samples, but the elucidation of their causal relationship to these diseases and the mechanisms involved is far from complete. In addition, some conflicting results that have been reported in this field further confirm the functional complexity of KChIPs. For example, the role of KChIP3 in pain appears to differ between rats and mice. Small-molecule compounds such as NS5806, a ligand for KChIP2, has different effects in different species or even different parts of the same species, which will significantly hamper the development of targeted therapies. Therefore, despite the development of several small molecules targeting KChIPs, further efforts are needed to demonstrate their translational value in treating disease.

## Figures and Tables

**Figure 1 cells-12-01894-f001:**
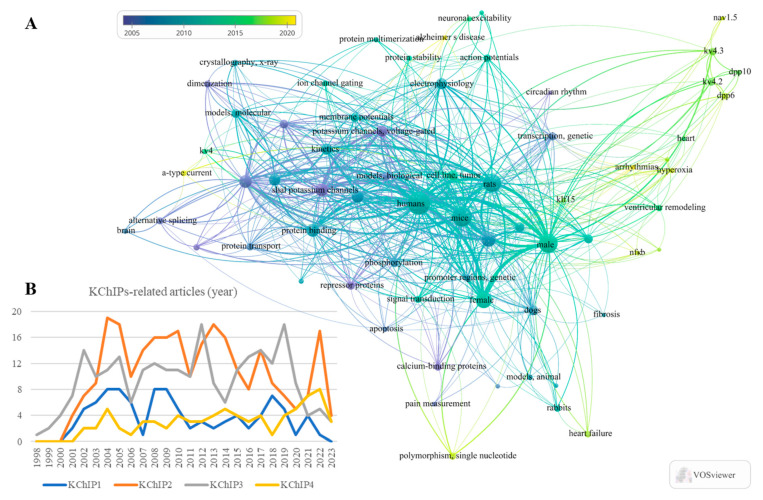
Statistics of the KChIPs-related publications and the network visualization of key words. (**A**) Bibliometrics information acquired by VOSviewer shows the key words that occurs frequently in these KChIPs-related publications. (**B**) The statistics of the KChIPs-related publications since the year 1998.

**Figure 2 cells-12-01894-f002:**
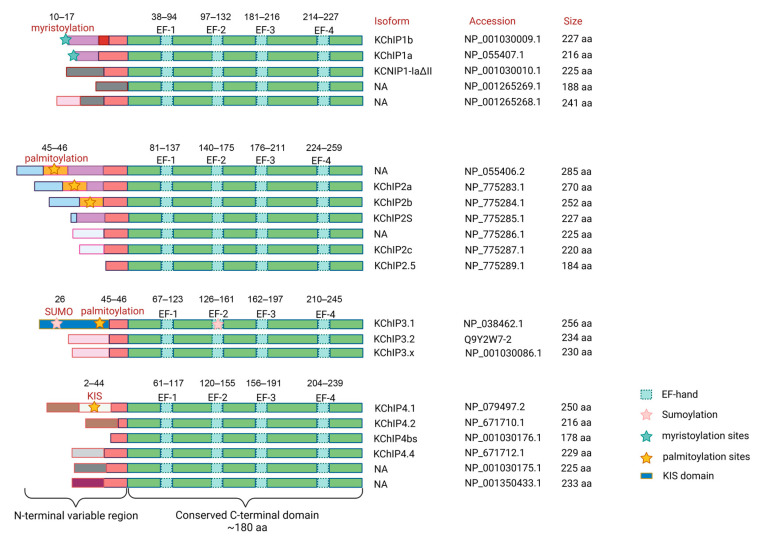
Diagram of the protein structure of the KChIP family. KChIPs are bipolar proteins of approximately 178–285 amino acids in size. Each KChIP contains an N-terminal domain and a C-terminal domain. The N-terminal domain differs between KChIPs and has multiple post-translational modification sites, whereas the C-terminal domain contains four conserved EF-hand Ca^2+^-binding motifs. Human *KCNIP1* has five transcript variants encoding five different isoforms (KChIP1a, KChIP1b, KChIP1-1adetall, and another two unnamed isoforms). *KCNIP2* has seven transcript variants and encodes seven different isoforms (KChIP2a, KChIP2b, KChIP2s, KChIP2c, KChIP2.5, and another two unnamed isoforms). *KCNIP3* has three transcript variants encoding three different isoforms (KChIP3.1, KChIP3.2, and KChIP3.x). *KCNIP4* has seven transcript variants encoding six different isoforms (KChIP4.1, KChIP4.2, KChIP4bs, KChIP4.4, and another two unnamed isoforms), of which variant 3 (NM_147182.4) and variant 6 (NM_001035004.2) encode the same isoform 3 (NP_001030176.1). Functional domains are indicated by boxes with different colors. The post-translational modification sites are marked with pentagrams of different colors. The corresponding protein names and accession numbers refer to NCBI or Uniprot database. NA, not available.

**Figure 3 cells-12-01894-f003:**
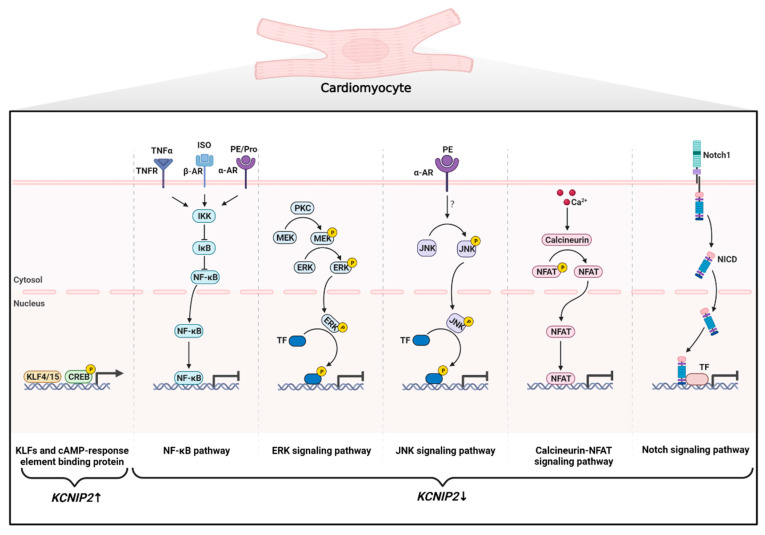
Regulation of KChIP2 expression at the transcriptional level in cardiomyocytes. Positive regulation of *KCNIP2* transcription includes: (1) members of the Krüppel-like factor family (KLF4 and KLF15) and (2) CREB directly binding to the promoter regions of the *KCNIP2* gene. Negative regulation of *KCNIP2* transcription includes: (1) Activation of TNFR, α-AR, and β-AR increases NF-κB activity and thus exerts an inhibitory effect on the *KCNIP2* promoter. (2) Two branches of the MAPK signaling pathway are involved in the downregulation of KChIP2 expression: Activation of PKC phosphorylates and activates MEK, which in turn activates ERK. JNK can be phosphorylated and activated by phenylephrine (PE). Both ERK and JNK can reduce *KCNIP2* mRNA levels by activating associated transcription factors. (3) The Ca^2+^-activated phosphatase calcineurin dephosphorylates NFAT, which then translocates to the nucleus to repress *KCNIP2* transcription. (4) The Notch1 receptor is cleaved by γ-secretase to generate NICD following receptor–ligand interaction. NICD then translocates to the nucleus and interacts with transcriptional regulators to inhibit *KCNIP2* expression. CREB: cAMP response element-binding protein; TNFR: tumor necrosis factor receptor; α-AR: α-adrenergic receptor; β-AR: β-adrenergic receptor; Pro: propranolol; NF-κB: nuclear factor κB; IκB: NF-κB inhibitor; IKK: IκB kinase; PKC: protein kinase C; MEK: mitogen-activated protein kinase kinase; ERK: extracellular regulated protein kinase; JNK: c-Jun N-terminal kinase; TF: Transcription factor; NFAT: nuclear factor of activated T cells; NICD: intracellular domain of the Notch receptor.

**Figure 4 cells-12-01894-f004:**
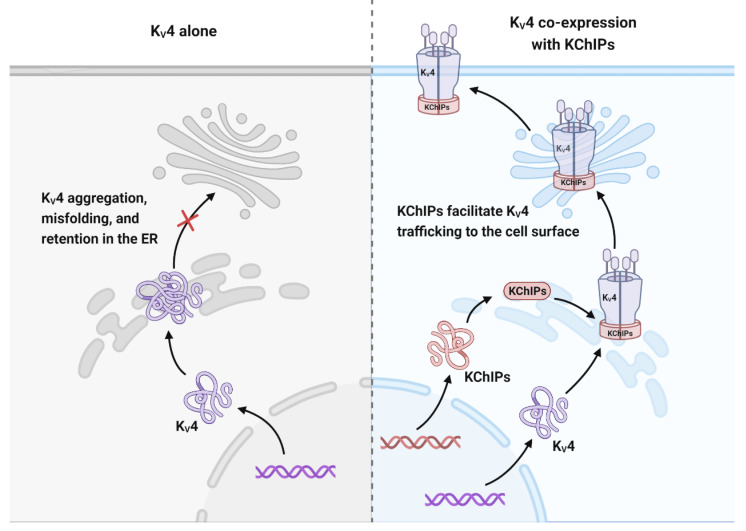
KChIPs allow K_V_4 trafficking to the cell surface. K_V_4 channels cannot be trafficked to the cell membrane on their own because they are aggregated, misfolded, and retained in the ER for degradation. Co-expression of KChIP1-3 releases K_V_4 channels from ER retention and redistributes them to the cell surface.

**Figure 5 cells-12-01894-f005:**
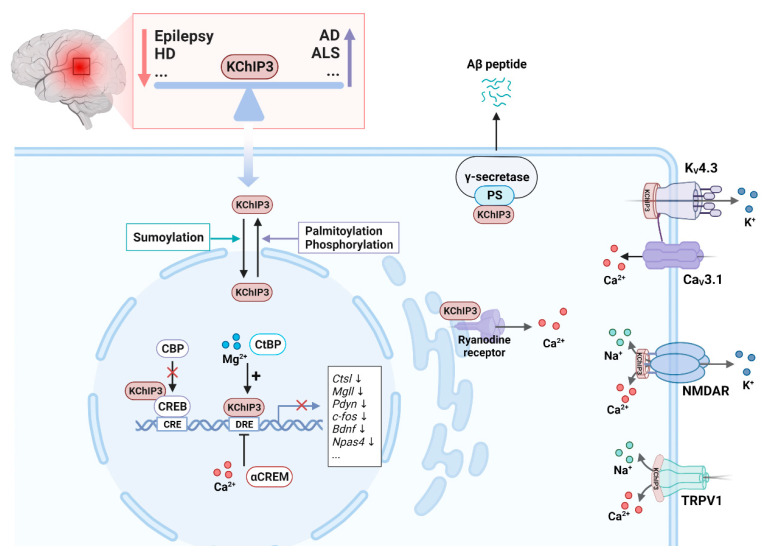
Function of KChIP3 in neurons. Altered KChIP3 expression is associated with several nervous system disorders. For example, KChIP3 is downregulated in epilepsy and Huntington’s disease (HD) and upregulated in Alzheimer’s disease (AD) and amyotrophic lateral sclerosis (ALS). KChIP3 has physiological functions in both the cytoplasm and the nucleus. Sumoylated KChIP3 translocates to the nucleus to repress the transcription of associated genes by interacting with the DRE sequences. This interaction can be enhanced by Mg^2+^ and C-terminal binding protein (CtBP) and inhibited by Ca^2+^ and cyclic AMP-responsive element modulator α (αCREM). KChIP3 can also repress target gene transcription by interacting with the transcription factor CREB to interfere with CREB phosphorylation and CREB-binding protein (CBP) recruitment. Palmitoylation and phosphorylation of KChIP3 increase its location in the cytoplasm, where it interacts with presenilin (PS) to regulate the enzymatic activity of the γ-secretase. In the cytoplasm, KChIP3 regulates intracellular K^+^, Ca^2+^, and Na^+^ currents by interacting with K_V_4, N-methyl-D-aspartate receptor (NMDAR), transient receptor potential vanilloid 1 (TRPV1), and the ryanodine receptor.

**Figure 6 cells-12-01894-f006:**
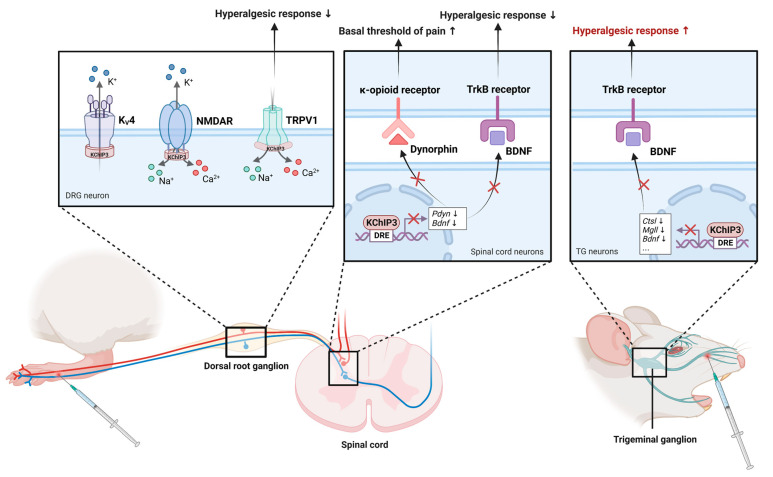
The role of KChIP3 in the modulation of pain. KChIP3 is involved in pain control in both the spinal cord/DRG (left) and the trigeminal ganglion (right). In DRG neurons, KChIP3 modulates pain transmission by directly interacting with K_V_4 channels, TRPV1 channels, and NMDAR. On the one hand, KChIP3 contributes to the central transmission of pain signals by increasing K_V_4 currents. On the other hand, KChIP3 exerts analgesic effects by inhibiting the surface expression of TRPV1 and NMDAR. In the spinal cord, KChIP3 directly binds the genes encoding dynorphin and BDNF and represses their expression. In trigeminal ganglion neurons, KChIP3 binds and inhibits *Bdnf*, *Ctsl*, and *Mgll* to regulate trigeminal noxious perception.

## Data Availability

Not applicable.

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
