# Peer review of "KV Channel-Interacting Proteins in the Neurological and Cardiovascular Systems: An Updated Review"

_cells, 2023, doi:10.3390/cells12141894_

Round 1

Reviewer 1 Report

The scientific  review on the role of KChIPS related to neurology and CVS  fits the scope of the  journal. It gives a clear ,elaborate , well structured update on the role of KV channel-interacting proteins in neurological and cardiovascular system which is relevant to the current scenario. Relevant recent publications have been added. No excessive self citation has been observed.

This review is a good scientific addition to the existing literature as there are very few recent reviews on this topic.

Overall the article is well written clearly highlighting the different roles of KV channel-interacting proteins highlighting both the promoting as well as the inhibiting role of the proteins.

The conclusion is succinct based on the given observations. The review article will be a good value addition to a wide range of scientists and researchers working on various aspects of neurological and cardiovascular disorders.

Author Response

We are grateful to the editor and reviewers for their constructive comments and suggestions, on the basis of which a number of revisions have been made to the manuscript. Below are our point-by-point responses to each comment.

Reviewer 1:

  1. The scientific review on the role of KChIPS related to neurology and CVS fits the scope of the journal. It gives a clear, elaborate, well structured update on the role of KV channel-interacting proteins in neurological and cardiovascular system which is relevant to the current scenario. Relevant recent publications have been added. No excessive self citation has been observed.
  2. This review is a good scientific addition to the existing literature as there are very few recent reviews on this topic.
  3. Overall the article is well written clearly highlighting the different roles of KV channel-interacting proteins highlighting both the promoting as well as the inhibiting role of the proteins.
  4. The conclusion is succinct based on the given observations. The review article will be a good value addition to a wide range of scientists and researchers working on various aspects of neurological and cardiovascular disorders.

Response: We are grateful for Reviewer #1’s generous and positive comments. Indeed, it is our intention to carry out a systematic review and supplementary update of the research progress on KChIPs from an objective point of view, in order to present a review that can be referenced by researchers in the field. Thank you again for your endorsement of our work.

Reviewer 2 Report

1. This review is updated, as can be seen from the many references published in the past three years. The topic is interesting, and can be helpful in future developments around KChIP. My major concern is the critical remarks. The authors pointed out the current limitations on the published studies, but they are expected to focus more on such critical comments in the whole manuscript. From the literature review in the introduction, the authors can show certain limitations. 

2. As a review, tables are more preferred to give directly more information, especially on the comparative analysis. 

3. One highlight of this paper is the "updated" review and analysis. Thus, curve figures are recommended to better illustrate the change of the research focus and trend in the past decade to show the updated progress. For example, the number of publications on certain topics changing with the time. 

4. More advanced models should be included to make this review more complete, for example, "https://doi.org/10.3390/ijms241210074". 

5. The concluding remarks should be strengthened. The authors should comment and conclude on the current development first, and then start point-to-point critical remarks. 

The English writing is fine. As a review paper, a nomenclature is recommended as many abbreviations and symbols are involved. 

Reviewer 3 Report

The paper proposed by Wu and colleagues is a very well written and organized review of the recent literature regarding KChIPs structure, functions and implications in diseases. Each part, except the last one, is illustrated by high quality and comprehensive figures. Data from a huge number of studies are reported, including the most recent.

I have two remarks which may improve the paper.

-          Models used in the reported studies are often missing. It could be interesting to indicate these models, when appropriate. As an example, the authors indicate that the “No DREM, no pain” does not apply to rats (line 474) but it is not clear, before, in which models it was proven.

-          The fourth chapter regarding KChiPs and diseases is somewhat complicated to read since it is a succession of individual studies. This is particularly true for the “Pain” chapter 4.1.2. In order to guide the reader, a summarizing figure or table relating KChiPs major roles in disease might be helpful.

Author Response

We are grateful to the editor and reviewers for their constructive comments and suggestions, on the basis of which a number of revisions have been made to the manuscript. Below are our point-by-point responses to each comment.

Reviewer 3:

The paper proposed by Wu and colleagues is a very well written and organized review of the recent literature regarding KChIPs structure, functions and implications in diseases. Each part, except the last one, is illustrated by high quality and comprehensive figures. Data from a huge number of studies are reported, including the most recent.

I have two remarks which may improve the paper.

  1. Models used in the reported studies are often missing. It could be interesting to indicate these models, when appropriate. As an example, the authors indicate that the “No DREM, no pain” does not apply to rats (line 474) but it is not clear, before, in which models it was proven.

Response: Thank you for highlighting this critical point. Indeed, the original version of this review failed to include the disease models used in the bibliographies, and this has no doubt confused the reader. We have therefore included the missing animal models in the revised manuscript, including the drug-induced pain models in mice and rats described in "Pain" Chapter 4.1.2. (please see page 15-17 in “Pain” chapter 4.1.2.). We also briefly introduced and described the daDREAM transgenic mice used in the reports (please see line 508-511 on page 14). We appreciate your valuable suggestions.

  1. The fourth chapter regarding KChIPs and diseases is somewhat complicated to read since it is a succession of individual studies. This is particularly true for the “Pain” chapter 4.1.2. In order to guide the reader, a summarizing figure or table relating KChIPs major roles in disease might be helpful.

Response: Thank you very much for this positive comment and valuable suggestion. The regulatory mechanism of KChIP3 in pain is extremely complex. We apologize for not making this part of the manuscript more fluent and understandable in the previous version. We have therefore enriched and optimized this part of the manuscript so that readers can better understand the research progress of KChIP3 in pain modulation. The modification we made is mainly to separate the description of the role of KChIP3 in the dorsal root ganglion and spinal cord of the pain pathway (please see page 15-17 in “Pain” chapter 4.1.2.). In addition, as suggested, we have now included a figure to summarize the roles in pain modulation (please see Figure 6 on page 15).

Figure 6. The role of KChIP3 in the modulation of pain. KChIP3 is involved in pain control in both the spinal cord/DRG (left) and the trigeminal ganglion (right). In DRG neurons, KChIP3 modulates pain transmission by directly interacting with KV4 channels, TRPV1 channels, and NMDAR. On the one hand, KChIP3 contributes to the central transmission of pain signals by increasing KV4 currents. On the other hand, KChIP3 exerts analgesic effects by inhibiting the surface expression of TRPV1 and NMDAR. In the spinal cord, KChIP3 directly binds the genes encoding dynorphin and BDNF and represses their expression. In trigeminal ganglion neurons, KChIP3 binds and inhibits Bdnf, Ctsl and Mgll to regulate trigeminal noxious perception.

Reviewer 4 Report

The manuscript was written by wu et al. entitled " KV channel-interacting proteins in neurological and cardiovascular system: An updated review ". This review focuses on the association between altered expression of KChIPs and various diseases such as arrhythmia, heart failure, and Alzheimer's disease. Researcher aim to summarize the latest research on the structural properties, physiological functions, and pathological roles of KChIPs in disease progression, while also discussing their potential as pharmacological targets for therapeutic purposes.

 Major comments:

1.       Definitely - the abstract should be significantly reworked and improved. This needs to be made more conceptual, more informative, and based more on the results of the theoretical research carried out, and especially of the novelty, originality and concrete contribution to the science.

2.      How do KChIPs modulate various aspects of channel function, such as cell surface trafficking, voltage-dependent activation, inactivation kinetics, and recovery rate from inactivation?

3.      Apart from their influence on KV4 channels, how do KChIPs regulate other ion channels like IKs, ICa,L, and INa?

4.      What is the significance of the predominant expression of KChIPs in the brain and heart, particularly in maintaining neuronal and cardiomyocyte excitability by modulating KV4 currents?

5.      Furthermore, what additional role do KChIPs serve as transcription factors, and how do they exert control over the expression of genes involved in pain, memory, and circadian regulation?

6.       How is the altered expression of KChIPs implicated in the development of diseases such as arrhythmia, heart failure, and Alzheimer's disease, and what does this suggest about their potential as contributors to disease pathogenesis?

7.      How does this review aim to enhance our understanding of the significance of KChIPs in disease mechanisms and offer potential avenues for therapeutic exploration?

Author Response

We are grateful to the editor and reviewers for their constructive comments and suggestions, on the basis of which a number of revisions have been made to the manuscript. Below are our point-by-point responses to each comment.

Reviewer 4:

The manuscript was written by wu et al. entitled " KV channel-interacting proteins in neurological and cardiovascular system: An updated review ". This review focuses on the association between altered expression of KChIPs and various diseases such as arrhythmia, heart failure, and Alzheimer's disease. Researcher aim to summarize the latest research on the structural properties, physiological functions, and pathological roles of KChIPs in disease progression, while also discussing their potential as pharmacological targets for therapeutic purposes.

 Major comments:

  1. Definitely - the abstract should be significantly reworked and improved. This needs to be made more conceptual, more informative, and based more on the results of the theoretical research carried out, and especially of the novelty, originality and concrete contribution to the science.

Response: Thank you for highlighting this important point. As suggested, we have modified the "Abstract" in the manuscript to make it more conceptual and informative. In addition, we have emphasized that this review is an update of the research progress of the "KChIPs" in which we summarized our views on the limitations of the current research, hoping to provide some reference for researchers in the field (please see "Abstract" on page 1).

  1. How do KChIPs modulate various aspects of channel function, such as cell surface trafficking, voltage-dependent activation, inactivation kinetics, and recovery rate from inactivation?

Response: Thank you for your critical question. In the revised abstract, we have added the mechanism of regulation of KV4 channels by KChIPs. We conclude that KChIP1, KChIP2 and KChIP3 promote the translocation of KV4 channels to the cell membrane and accelerate voltage-dependent activation and slow the recovery rate of inactivation (please see lines 14-19 on page 1). These mechanisms are described in detail in "KChIPs modulate the gating properties of KV4 channels" in section 3.1.2 on page 7-8 and "KChIPs modulate the trafficking of KV4 channels" in section 3.1.3 on page 8. Your opinion is very important for us to improve the manuscript, thank you again for your valuable feedback.

  1. Apart from their influence on KV4 channels, how do KChIPs regulate other ion channels like IKs, ICa,L, and INa?

Response: Thanks for your critical question. In the revised “Abstract”, we have added the mechanism of regulation of IKs, ICa,L and INa by KChIPs. We conclude that ICa,L and INa are positively regulated by KChIP2, whereas IKs is negatively regulated by KChIP2 in the heart (please see lines 19-21 on page 1). These mechanisms are described in detail in "Role of KChIPs in regulating other ion channels" in section 3.2 on page 11. Once again, thank you for your valuable feedback.

  1. What is the significance of the predominant expression of KChIPs in the brain and heart, particularly in maintaining neuronal and cardiomyocyte excitability by modulating KV4 currents?

Response: Thank you very much for your critical question. KChIPs are predominantly expressed in the brain and heart, where they help maintain the excitability of neurons and cardiomyocytes by modulating fast inactivating KV4 currents. And in the absence of KChIPs, the suspected ability to cause epilepsy and arrhythmias has been increased in animal models. Due to space limitations in the “Abstract”, we have not described these in detail here. But support for this view can be found in paragraph 1 of "The interaction of KChIPs with KV4 channels" in section 3.1.1 on page 6, "Epilepsy" in section 4.1.1 on page 13 and "Arrhythmias" in section 4.2.1 on page 18. Your opinion is very important for us to improve the manuscript, thank you again for your valuable feedback.

  1. Furthermore, what additional role do KChIPs serve as transcription factors, and how do they exert control over the expression of genes involved in pain, memory, and circadian regulation?

Response: Thanks for your critical question. All KChIPs have Ca2+-dependent DRE binding affinities and are able to block transcription of target genes containing DRE sites. KChIP3 has been reported to regulate several genes involved in pain modulation and memory, such as Pdyn, Bdnf, Mgll, Ctsl, c-fos and Npas4. KChIP1-4 are involved in the regulation of rhythmically expressed genes involved in circadian rhythms, including Aanat, Crem and Fra-2. Detailed descriptions can be found in lines 397-404 on page 11, lines 432-436 on page 12 and paragraph 2 in "Pain" chapter 4.1.2. on page 14. Thank you again for your valuable feedback.

  1. How is the altered expression of KChIPs implicated in the development of diseases such as arrhythmia, heart failure, and Alzheimer's disease, and what does this suggest about their potential as contributors to disease pathogenesis?

Response: Thank you for your critical question. KChIP2 is decreased in failing hearts, while loss of KChIP2 leads to increased susceptibility to arrhythmias. In addition, KChIP3 is increased in Alzheimer's disease and amyotrophic lateral sclerosis, but decreased in epilepsy and Huntington's disease. This section has been added to the newly revised manuscript. Detailed descriptions can be found in "Alzheimer's disease" section 4.1.4. on page 17, "Arrhythmia" section 4.2.1. on page 18, and "Heart failure" section 4.2.3. on page 19. Your opinion is very important for us to improve the manuscript, thank you again for your valuable feedback.

  1. How does this review aim to enhance our understanding of the significance of KChIPs in disease mechanisms and offer potential avenues for therapeutic exploration?

Response: Thank you for your critical comment. In this review, we summarized the progress of recent research on the structural properties, physiological functions and pathological roles of KChIPs in health and disease. We also summarized the small molecule compounds that regulate the function of KChIPs, which are potential avenues for therapeutic exploration (please see Table 2 in page line). Additionally, we discussed the controversial observations of this area and gave potential explanations about them. We believe this review provided an overview and update of the regulatory mechanism of the KChIP family and the progress of targeted drug research, as a reference for researchers in related fields. Once again, thank you for your valuable feedback.

Round 2

Reviewer 4 Report

well improved, kindly accept it

Author Response

Response: Thank you for your recognition. It means a lot to us.